# The Nucleoshuttling of the ATM Protein: A Unified Model to Describe the Individual Response to High- and Low-Dose of Radiation?

**DOI:** 10.3390/cancers11070905

**Published:** 2019-06-28

**Authors:** Elise Berthel, Nicolas Foray, Mélanie L. Ferlazzo

**Affiliations:** Institut National de la Santé et de la Recherche Médicale, UA8, Radiations: Defense, Health and Environment, Centre Léon-Bérard, 28, rue Laennec, 69008 Lyon, France

**Keywords:** radiosensitivity, ATM, DSB repair

## Abstract

The evaluation of radiation-induced (RI) risks is of medical, scientific, and societal interest. However, despite considerable efforts, there is neither consensual mechanistic models nor predictive assays for describing the three major RI effects, namely radiosensitivity, radiosusceptibility, and radiodegeneration. Interestingly, the ataxia telangiectasia mutated (ATM) protein is a major stress response factor involved in the DNA repair and signaling that appears upstream most of pathways involved in the three precited RI effects. The rate of the RI ATM nucleoshuttling (RIANS) was shown to be a good predictor of radiosensitivity. In the frame of the RIANS model, irradiation triggers the monomerization of cytoplasmic ATM dimers, which allows ATM monomers to diffuse in nucleus. The nuclear ATM monomers phosphorylate the H2AX histones, which triggers the recognition of DNA double-strand breaks and their repair. The RIANS model has made it possible to define three subgroups of radiosensitivity and provided a relevant explanation for the radiosensitivity observed in syndromes caused by mutated cytoplasmic proteins. Interestingly, hyper-radiosensitivity to a low dose and adaptive response phenomena may be also explained by the RIANS model. In this review, the relevance of the RIANS model to describe several features of the individual response to radiation was discussed.

## 1. Introduction

To date, the evaluation of the risks linked to an exposure to radiation, whether clinical, occupational, or environmental, has become a societal, medical, and scientific issue. This is notably the case of the investigations about the secondary effects of anti-cancer radiotherapy [1,2], the radiation-induced cancers after repeated mammographic views in young women [3], and the radiation-induced (RI) pathologies observed in nuclear workers [4]. Unfortunately, there is still no consensus about the prediction of these RI risks from molecular and cellular data [2]. This review is therefore devoted to the quest of a unified mechanistic model that would be the basis of a reliable prediction of the individual response to radiation.

Less than 10 years after the discovery of X-ray by Roentgen [5], the three major clinical consequences of irradiation were identified already (Figure 1):Radiosensitivity responses, i.e., adverse tissue events, are non-cancer effects, attributable to cell death. First reported by Giezel, Voigt, Albers-Schönberg, and Bouchacourt [6,7,8], detailed descriptions of radiodermatitis and RI reactions to other irradiated organs have progressively led to the definition of consensual severity scales [2,9], like the Common Terminology Criteria for Adverse Events (CTCAE) [10] and the Radiation Therapy Oncology Group (RTOG) [11] scales. These two scales classify RI tissue reactions in six grades (grade 0: no event; grade 5: death), for each organ and independently of the early/late nature of the reactions. The CTCAE or RTOG severity grades are the most reliable endpoints to quantify clinical radiosensitivity. On a biological scale, the quantification of radiosensitivity is dependent on the whole knowledge of RI cell death. The only consensual endpoint to quantify cellular radiosensitivity is clonogenic cell survival, which obeys the empirical linear-quadratic (LQ) model [12,13,14]. However, the cell survival assays are too time-consuming to be applicable in routine. Lastly, while the skin burns and other RI tissue reactions were described earlier, it is noteworthy that the term “radiosensitivity” appeared for the first time in 1907 [15].Radiosusceptibility responses, i.e., RI cancers, are non-toxic effects attributable to cell transformation and genomic instability. First reported in 1902 [16], and revealed to the public by the story of the radium dial painters [17], RI cancers have been significantly documented by the reports of Hiroshima survivors [18,19,20]. To date, the most reliable endpoints to quantify the risk of RI cancers is the excess relative risk ratio (ERR) or any related endpoints from epidemiology [21]. However, the statistical robustness of these endpoints is strongly dependent on the size of the cohorts studied. To describe the ERR as a function of radiation dose, two major models were proposed: the linear non-threshold (LNT) and the non-linear threshold (NLT) models. The relevance of these two empirical models is still the source of controversy [22,23,24]. On a biological scale, the quantification of radiosusceptibility is dependent on basic knowledge about carcinogenicity mechanisms. To date, the numbers of G2 chromosomal aberrations [25] and hypoxanthine phosphoribosyltransferase (*HRPT*) mutations frequency [26] may be considered as the most specific endpoints of the RI cellular transformation but are not consensual. Lastly, while the first RI cancers were described earlier, it is noteworthy that the term “radiosusceptibility” was proposed for the first time in 2016, to avoid any confusion with the use of “radiosensitivity” [15].Radiodegeneration responses, i.e., non-cancer effects, are non-cancer effects attributable to mechanisms related to accelerated aging [18,27]. First reported in 1903 in humans, RI cataracts are the most frequent radiodegeneration response [28]. RI cardiovascular effects, first reported in 1932, also belong to this category [29]. Like for RI cancers, the estimation of the incidence of RI radiodegeneration effects is limited by the lack of specific epidemiological data. Similarly, on a biological scale, the quantification of radiodegeneration is dependent on basic knowledge of senescence and aging mechanisms. Telomere length and telomerase activity are frequently cited as the most specific endpoints to describe aging [2,30]. Lastly, it is noteworthy that the term “radiodegeneration” was proposed for the first time in 2016, in order to distinguish syndromes associated with cancer proneness and those associated with aging [15].

Although the three major RI effects differ by their clinical features and their molecular origins, they share common points. Particularly, they obey specific dose-, time-, and dose-rate effect functions that are not necessarily linear and can present some thresholds [2,31]. Such dependence on both dose and time suggests that functional assays (i.e., those involving non-irradiated and irradiated cells) should be more suited to predicting the three major RI effects than the approaches derived from spontaneous data only (like DNA sequence, genomics, proteomics on non-irradiated cells). In addition to this dose- and time- dependence, the occurrence and the degree of the three precited RI effects strongly depend on the individual status [2] and on the irradiated tissues/organs [27]. The prediction of individual radiation response therefore requires a wide spectrum of individuals/tissues to be tested in order to be reliable statistically [32].

## 2. A Survey of Human Radiosensitivity

Among the three RI effects mentioned above, individual radiosensitivity is historically the most documented [15]. Hence, let us focus on this notion through its different clinical features and the different attempts to predict them. 

### 2.1. The Different Clinical Features of Radiosensitivity

By omitting accidental irradiation, radiosensitive individuals can be divided into two subcategories. The first one gathers individuals who have been treated against cancer by radiotherapy and who showed RI tissue reactions with CTCAE/RTOG grades higher than 1. This subpopulation may represent 5–20% of individuals [2]. With the exception of well-characterized genetic diseases (see below), these individuals did not show obvious and specific clinical signs of radiosensitivity before their treatment. The second subcategory gathers individuals who suffered from a well-characterized genetic disease for which radiosensitivity is one of the numerous symptoms. In these cases, the radiosensitivity can be revealed during a radiotherapy treatment as far as the disease is also associated with cancer proneness. In the other cases, generally related to neurodegenerative diseases that are not associated with high cancer proneness (like progeria [33] or Huntington’s disease [34]), the radiosensitivity is revealed by in vitro radiobiological studies on cells from patients. It is noteworthy that the subcategory defined above can also concern patients whose genetic disease has not been diagnosed [35,36].

The relationship between clonogenic cell survival and radiosensitivity has been considerably documented [37] and surveys of human radiosensitivity make it possible to present a general picture of human cellular radiosensitivity, encountered in about 30 genetic diseases [38,39]. The survival fraction at 2 Gy (SF2) appears to be one of the best parameters to quantify cellular radiosensitivity and in good agreement with the clinical response to radiation [39]. In human non-transformed fibroblasts, SF2 ranges from 1–70% [39,40] (Figure 2):
Hyper-radiosensitivity: The most hyper-radiosensitive cells (SF2 ranging from 1–10%) derive from leukemia/lymphoma patients suffering from homozygous mutations of the Ataxia Telangiectasia Mutated (*ATM)* gene (the highest hyper-radiosensitivity observed in humans) and homozygous mutations of the ligase IV (*LIG4*) gene (only one case reported) who succumbed after radiotherapy or homozygous mutations of the Nijmegen Breakage Syndrome (*NBS1*) gene. Furthermore, the mutations of lamina A (*LMNA*) derived from patients suffering from the progeroid Hutchinson–Gilford syndrome belong to this group [2,33,39,41,42]. The cumulative incidence of these syndromes does not exceed 1%: they represent, therefore, a minority of patients, whose symptoms are mostly detectable in pediatrics. On the biological scale, all these mutations result in the loss of protein function and lead to a strong inhibition of DNA double-strand breaks (DSB) recognition or repair [2,33,39,41,42].Moderate radiosensitivity: SF2 ranging from 10–50% corresponds to a moderate sensitivity, such as that observed in genetic syndromes associated with high cancer proneness, like Fanconi anemia (*FANC* mutations), Bloom’s syndrome (*BLM* mutations), and neurofibromatosis (*NF1* mutations). Another subset of genetic syndromes belonging to this subcategory gathers aging and/or neurodegenerative diseases like Cockayne syndrome (*CS* mutations) or Huntington’s disease (*HTT* mutations) [2,39]. Such moderate radiosensitivities do not correspond to fatal reactions after radiotherapy but to morbidity reactions (i.e., CTCAE/RTOG severity grade ranging from 2 to 4). The cumulative incidence of the cases of moderate radiosensitivity represents the majority of patients who showed significant post-radiotherapy tissue reactions [2]. At the biological scale, all these mutations do not necessarily result in the loss of protein function but lead to a relative inhibition of DSB repair and signaling. Furthermore, it is noteworthy that some heterozygous mutations are associated with an overexpression of the mutated protein, like with Li Fraumeni syndrome (heterozygous *p53* mutations) [43].Normosensitivity (or radioresistance): SF2 ranging from 50–70%, even up to 80% for some tumors, corresponds to individuals considered “radioresistant”, who do not suffer from cancer (with the notable exception of occupational cancers) and who do not show any secondary effects after radiotherapy (CTCAE/RTOG grade 0) [2]. Normosensitivity is often defined by historical cell lines, for which patient follow-up is well characterized. However, normosensitive controls are difficult to obtain since a patient may or may not show post-radiotherapy tissue reactions, according to the radiotherapy modality and the way of delivering the dose [35].

Interestingly, some of the syndromes that are associated with a moderate radiosensitivity [39] are caused by mutations of cytoplasmic proteins, like Huntington’s disease [34], neurofibromatosis type I [44], tuberous sclerosis [45], and Bruton’s [46] and Usher’s syndromes [47]. This is also the case of the progeroid Hutchinson–Gilford syndrome, caused by mutations of lamina A, that is not directly involved in DNA damage signaling and repair [33]. Such observations suggest that radiosensitivity is not necessarily based on DNA repair defects only and raise questions about the importance of the nuclear membrane permeability and radiosensitivity. The radiosensitivity observed in syndromes caused by cytoplasmic mutated proteins can be therefore considered an enigma of radiobiology. 

### 2.2. The Major Approaches to Predict Radiosensitivity and Their Limits

To date, the major approaches to predict radiosensitivity do not reach all the above requirements for a reliable prediction of radiosensitivity: Assays based on cellular death: while SF2 is one of the best parameters to quantify cellular radiosensitivity [39], clonogenic cell survival assays are too time-consuming to predict radiosensitivity in routine. Assays based only on a particular cell death are not robust enough statistically to reliably predict radiobiology [32,48]. For example, assays based on apoptosis are irrelevant for predicting the radiosensitivity of fibroblasts that do not show this type of cell death. Furthermore, when applied on lymphocytes, apoptotic assays provide an inverse correlation between apoptotic yield and clinical radiosensitivity (the higher the apoptotic yield, the more radioresistant the patient is) which is not in agreement with the current models and needs further investigation [2,49].Assays based on cytogenetics: yields of unrepaired chromosomes, and especially micronuclei, have been quantitatively correlated with radiosensitivity [2,50]. However, the ranges of unrepaired chromosomes and of micronuclei are too small (0–12% and 0–25% per 100 cells, respectively) to reflect moderate radiosensitivity. The predictive power of cytogenetic endpoints is therefore limited [35].Assays based on DSB repair: while there is a quantitative correlation between unrepaired DSB and SF2, such a correlation does not make it possible to predict the intermediate radiosensitivity, for the same reasons evoked above with cytogenetics: the yield of unrepaired DSB ranges between 0 and 8 while SF2 varies from 1–70% [35,40].Genomics: as evoked above, the boolean nature (yes/no) of the DNA sequence endpoints cannot account for any dose-function. For example, any endpoint from genomics cannot provide biological interpretation of the LQ model. Conversely, genomics data are very useful for identifying gene mutations and new syndromes associated with radiosensitivity [51].

## 3. ATM, a Nucleocytoplasmic Protein Upstream of the Molecular Response to Radiation

### 3.1. ATM, a Nucleocytoplasmic Protein Early Activated after Irradiation

Since the 1970s, ataxia telangiectasia caused by homozygous mutations of ATM has been known to be associated with the highest radiosensitivity encountered in humans [52]. Since the SF2 of the *ATM*-mutated cells is systematically lower than that of the other radiosensitivity cases, the ATM protein is likely to be upstream of the molecular process of the radiation response [53]. Interestingly, the SF2 of the *ATM*-mutated cells is systematically lower than that of the *LIG4*-mutated cells, suggesting that the loss of ATM activity itself leads to more severe consequences, such as a strong defect in the non-homologous end-joining pathway (NHEJ) (the predominant DSB repair pathway in humans) [19]. Hence, ATM may act upstream of the NHEJ pathway. In addition to these observations, it was suggested that ionizing radiation and oxidative stress trigger the ATM monomerization, which stimulates the ATM kinase activity [54]. Because of the current hypothesis that DNA damage is the origin of any cellular radiosensitivity, the ATM protein has long been considered as mainly nuclear, even if there was no clear evidence of the absence of cytoplasmic ATM forms. By contrast, to date, there is increasing evidence that ATM kinase is also a cytoplasmic protein. In 1998, Oka and Takashima and Lim. et al. were the first groups to evoke the cytoplasmic form of ATM [55,56]. Furthermore, a number of reports provided solid evidence of the existence of cytoplasmic forms of ATM but also of a nucleoshuttling of ATM with different experimental approaches, like immunofluorescence, immunoprecipitation, and enzyme-linked immunosorbent assay (ELISA) techniques [57,58,59,60,61,62].

### 3.2. ATM and the Other Serine/Threonine Kinases Involved in the DNA Damage Recognition

In parallel, the ATM-dependent phosphorylation of the variant histone H2AX on serine 139 (γH2AX) was shown to be one of the earliest RI events of the NHEJ pathway, through its relocalization as discrete nuclear foci, easily quantifiable by immunofluorescence [63]. The formation of γH2AX foci was considered to be the recognition step of NHEJ [63]. It is noteworthy that the γH2AX foci were also shown to be produced by Ataxia Telangiectasia mutated and RAD3-related (ATR) and DNA-dependent protein kinase (DNA-PK) kinases after genotoxic stress [64,65]. However, the roles of these two proteins in radiosensitivity seem to be different. Firstly, the mutations of the *ATR* gene (that notably cause Seckel’s syndrome) do not lead to the same level of hyper-radiosensitivity as that observed in *ATM*-mutated cells [66,67]. Furthermore, after exposure to UV, the ATR kinase is activated early post-exposure while ATM is not activated. By contrast, after exposure to ionizing radiation, ATM is activated in the first minutes post-irradiation, while ATR is activated after the first hour post-irradiation [53]. Hence, the γH2AX foci that could be potentially produced by ATR have been observed much later that those produced by ATM [53,64]. With regard to DNA-PK, a heterozygous mutation of the *DNA-PK* catalytic subunit (DNA-PKcs) gene has been found in a patient suffering from severe combined immunodeficiency (RS-SCID). However, this mutation does not affect DNA-PK activity and this syndrome is not associated with hyper-radiosensitivity like *ATM*-mutated patients [67,68]. There is no human syndrome associated with homozygous *DNA-PK* mutations, probably because they cause embryonic lethality [68]. Furthermore, the *DNA-PK*-mutated M059J tumor cells show an early formation of γH2AX foci (in the first hour post-irradiation), with a number similar to that assessed in radioresistant cells. By contrast, in the same *DNA-PK*-mutated tumor cells, the number of residual γH2AX foci reflecting unrepaired DSB was found to be persistent from 4 h post-irradiation, like in *LIG4*-mutated cells, suggesting a strong defect in NHEJ [40,69]. Altogether, these data show that despite some significant influence in the RI DNA damage repair and signaling, the mutations of the ATR and DNA-PK kinases taken separately is of a lesser extent than those of ATM and the level of their activity cannot explain the large spectrum of human radiosensitivity.

Finally, it was also suggested that some foci formed by γH2AX and tumor suppressor p53 binding protein 1 (53BP1) co-localize on the DSB sites. However, the exact temporal co-localization was not observed in the first 1 h post-irradiation [70]. For example, 10 min after 2 Gy, only 15 53BP1 foci per Gy per cell were observed in radioresistant fibroblasts, while about 80 were scored with γH2AX antibodies [40]. No correlation was established between radiosensitivity and 53BP1 foci formation, suggesting that co-localizations with γH2AX foci should be interpreted with caution [70]. Lastly, it must be stressed that all of the data mentioned here are related to quiescent cells and not proliferating cells. Since 53BP1 foci have been observed in S-G2/M cells, differences in the 53BP1 action vis-à-vis the cell cycle phase should also be considered [70]. Altogether, these data suggest that the number of 53BP1 foci cannot reliably predict human radiosensitivity. 

### 3.3. A Crucial Observation Raising Basic Questions about the Role of ATM

In the frame of our systematic study of human radiosensitivity, by applying γH2AX immunofluorescence on fibroblasts from hundreds of patients showing a wide range of post-radiotherapy radiosensitivity, the number of γH2AX foci assessed 10 min after radiation was found to be systematically lower than the expected induction rate of about 40 γH2AX foci per Gy per cell [35] (Table 1). Besides this, since ATM phosphorylates H2AX to form γH2AX foci, the γH2AX foci are rare or absent in the *ATM*-mutated cells. Conversely, the number of γH2AX foci in the *LIG4*-mutated cells reaches the expected values of about 40 γH2AX foci per Gy per cell, (similar to normosensitive cells) since the *LIG4*-mutated cells show a normal ATM function but impaired NHEJ activity [40]. All the literature data converge to the double statement that the DSB induction is a physico–chemical but not a biological process and that irradiation induces the same number of DSB per Gy per cell, independent of radiosensitivity [63,71]. Conversely, a lower number of γH2AX foci and a moderate cellular radiosensitivity suggest decreased ATM kinase activity in the nucleus [35]. When applying pATM immunofluorescence to the same cells, the number of pATM foci appeared also to be lower in cells showing a lower number of γH2AX foci [35] (Table 1). It was therefore hypothesized that the diffusion of ATM from the cytoplasm to the nucleus was impaired, which may explain the lower ATM kinase activity in the nucleus. 

## 4. The RIANS Model: A Solid Basis for Predicting Radiosensitivity

### 4.1. Major Principles of the RIANS Model

From the observations described above and derived from our COPERNIC collection of more than 100 fibroblast cell lines, the RIANS model was therefore proposed [35,71] (Figure 3). The two major hypotheses are that ATM is mainly situated in cytoplasm as dimers formed by two autophosphorylated (pATM) monomers at serine 1981, and that ionizing radiation triggers the monomerization of the cytoplasmic ATM dimers in a dose-dependent manner, as suggested already in the literature [72,73]. Thereafter, the ATM monomers, that are active forms, diffuse in the nucleus, probably more easily than dimers for steric reasons [71]. Active ATM monomers phosphorylate H2AX, which triggers NHEJ, and phosphorylate some nucleases, like MRE11, which inactivates the MRE11-dependent recombination-like DSB repair pathway responsible for cancer proneness or genomic instability [71]. Once DSBs are repaired, the proximity of the two active ATM monomers helps in forming a dimer, which produces pATM nuclear foci. This last step is supported by a ratio of 2, observed routinely between the number of early γH2AX foci and the early pATM foci, as shown previously [74]. Any delay of RIANS therefore leads to radiosensitivity and/or genomic instability, as validated by the COPERNIC fibroblasts collection [35]. The hyper-radiosensitivity of *ATM*-mutated cells is naturally explained by the absence of an ATM kinase activity in the nucleus (no DSBs are recognized by NHEJ), while that of *LIG4*-mutated *ATM*-mutated cells is explained by a gross repair defect by NHEJ (all the DSBs are recognized but they are not repaired by NHEJ) [71]. At this stage, to our knowledge, there is no other relevant model that could qualitatively and quantitatively explain the individual radiation response by considering ATM expression or other phophosphorylated or steric forms of ATM. However, since some encouraging studies suggest that ATM expression may serve as a prognostic factor in medical oncology [75], further investigation is still needed to relate the kinase activity of ATM with its expression level in both healthy and tumor cells. 

### 4.2. A Reliable Prediction of Individual Radiosensitivity

From the same COPERNIC fibroblasts collection, a quantitative correlation between CTCAE/RTOG severity grades (reflecting clinical radiosensitivity) and the number of the pATM foci (reflecting cellular radiosensitivity) was observed [35]. This correlation was found to be independent of the early/late nature of the post-radiotherapy reactions and of the nature/localization of the tumor [35]. Such a correlation is at the basis of a reliable prediction of clinical radiosensitivity from skin biopsies sampled before radiotherapy. The clinical applicability of the pATM assay was demonstrated in 2016 [35,76]. A variant pATM assay, based on the ELISA technique, was proposed in 2018. This faster pATM assay has the advantage of avoiding the time-consuming cellular proliferation step, but provides a lower statistical robustness than the pATM immunofluorescence assay [77]. However, despite these differences, both pATM assays (via immunofluorescence or ELISA) were shown to be characterized by the highest statistical performances among all the predictive assays proposed in the literature [32,49,77]. Hence, the best endpoints to reflect the RIANS remain the number of nuclear pATM foci observed by immunofluorescence and, to a lesser extent, the number of nuclear pATM molecules observed by the ELISA technique.

### 4.3. Three Groups of Human Radiosensitivity

From our data, the analysis of the delay in the RIANS in human cells enables us to define three groups of radiosensitivity [32,35,71,76,77,78] (Figure 3): Group I (about 75–85% of the whole population) represents the normosensitive (radioresistant) patients with a rapid RIANS after 2 Gy, and a low risk of post-radiotherapy tissue reaction and cancer;Group II (about 5–20% of the whole population) represents the patients who elicit a delay in the RIANS because of the sequestration of ATM in cytoplasm due to the formation of new cytoplasmic ATM substrates or their overexpression. These patients are moderately radiosensitive and susceptible to either cancer or to neurodegenerative disease;Group III (<1% of the whole population) represents the *ATM*-mutated patients or those who show strong DSB repair defects, hyper-radiosensitivity, and either high cancer proneness or severe accelerated aging [40,79].

Such classification is in good agreement with the observations described above in chapter 2. Indeed, for the homozygous mutations or some neo (mosaicism) mutations of proteins essential for the vital cellular functions, a significant dysfunction is observed (group III cases), but such gene mutations remain very rare. For the heterozygous gene mutations that are more frequent, a paradoxical overexpression of the mutated protein may contribute to delaying the RIANS by a more probable binding to ATM in the cytoplasm (group II cases) [2,34,45].

### 4.4. Radiosensitivity Caused by Mutated Cytoplasmic Proteins

One of the most important success of the RIANS model is the resolution of the radiosensitivity caused by mutated cytoplasmic proteins. For example, Huntington’s disease is associated with cellular radiosensitivity, while the huntingtin is a cytoplasmic protein. Mutations of huntingtin lead to the fusion of huntingtin with polyQ substrates. The wild-type huntingtin is not an ATM substrate. By contrast, ATM may bind to the polyQ domain of the mutated huntingtin, which may favor a binding of ATM to huntingtin in the cytoplasm and result in delaying the RIANS [34,80]. Another example is given by tuberous sclerosis complex (TSC) syndrome. TSC is caused by mutations of either the hamartin or tuberin proteins, which are mainly cytoplasmic. In the cytoplasm, ATM interacts with the TSC complex, which contributes to inhibiting the mTOR pathway. In the case of heterozygous mutations of the TSC2 protein, the TSC1 and TSC2 proteins separate, which contributes to activating the mTOR pathway. In parallel, the over-expression of the TSC2 protein contributes to sequestrating ATM in the cytoplasm [45]. In addition, neurofibromatosis type 1 (NF1) is also affected by these characteristics. NF1 is caused by heterozygous mutations of neurofibromin, a cytoplasmic protein that is an ATM substrate. Since these mutations are also associated with overexpression, they favor the ATM–neurofibromin complex in the cytoplasm, which results in a sequestration of ATM and a delay in the RIANS [44]. Similar observations in cells from patients suffering from retinoblastoma, Bruton’s tyrosine kinas, and Usher’s syndrome, all caused by cytoplasmic proteins, are in progress or have been submitted (work in progress, N.F. personal communication). Altogether, these data consolidate the relevance of the RIANS model but also suggest that some other proteins and mechanisms are to be identified to explain how the same delay in the RIANS may lead to different clinical syndromes. It is noteworthy that, to our knowledge, there is no other unified mechanistic model that provides a common molecular explanation to the radiosensitivity of these specific syndromes.

## 5. A Unified Model to Describe the Response to High- and Low-Dose of Radiation?

### 5.1. A New Biological Interpretation of the LQ Model

For nearly one century, the radiation response has been described by the target theory. In those studies, cell survival was described by a Poisson distribution of the lethal DNA damage [14]. In the frame of the RIANS model, two types of lethal DSB have been hypothesized: (1) The α-type DSBs, which are recognized by the ATM monomers in the nucleus (presence of γH2AX foci) but which remain unrepairable (persistent γH2AX foci). The number of the α-type DSBs, N_α_, was demonstrated to be proportional to the radiation dose D with α, as the proportionality coefficient: N_α_ = α D [71]; (2) The β-type DSBs, which are not recognized by the ATM monomers in the nucleus because of a delay or an absence of the RIANS (absence of γH2AX foci). The number of β-type DSBs, N_β_, was demonstrated to be proportional to the square of the radiation dose D with β, as the proportionality coefficient: N_β_ = β D^2^ [71]. As a result, the sum N(D) = N_α_(D)+ N_β_(D) represents the number of all lethal DSBs as a function of the dose, whatever the origin of the lethality of the unrepaired DSB. The expression of the clonogenic cell survival S as a function of dose therefore becomes: S = exp(−N(D)), i.e., corresponding to the formula of the LQ model S = exp(−α D − β D^2^) [71]. For the first time, to our knowledge, the LQ (α, β) parameters received a relevant biological interpretation that links the RIANS model to cell survival and to the capacity of the cell to mobilize the ATM kinase activity in the nucleus [71].

### 5.2. A Relevant Explanation for the Hyper-Radiosensitivity of the Low Dose Phenomenon?

The hyper-radiosensitivity to the low dose phenomenon (HRS) results in a significant reduction (about 25%) in clonogenic cell survival, an increase in the number of chromosome breaks, micronuclei, and unrepaired DSB between 1 and 500 mGy [81,82,83], which represents a biological effect equivalent of a dose 5 to 10 times higher [83,84]. The HRS phenomenon also concerns cellular transformation, since HRS has been observed with hypoxanthine-guanine phosphorybosyltransferase (HPRT) gene mutations as an endpoint [85]. The HRS phenomenon has been observed in numerous types of cells, whether derived from healthy tissues or tumors, but more preferentially in proliferating cells and in cells showing moderate radiosensitivity (see also below) [81,82,83]. Despite of the number of hypotheses, the intrinsic mechanisms of HRS remain unsolved [83].

The induction rate of the RI DSB assessed experimentally in a number of cells [63] and the induction rate of the RI ATM monomers, as suggested by literature data [72,73] and simulated in the frame of RIANS model [71], depend on the radiation dose but not on individual radiosensitivity. They were found to be about 40 RI DSB and 10,000 RI ATM monomers per Gy per human fibroblast, respectively [71]. By contrast, the number of ATM monomers that diffuse in the nucleus is strongly dependent on the RIANS and therefore on the individual radiosensitivity group. Hence, if all the RI DSB are recognized 10 min after 2 Gy in radioresistant (group I) cells, this is not the case for moderately radiosensitive (group II) cells. At low doses, the number of RI DSB and ATM monomers is reduced—the radiation dose may not be high enough to produce a significant amount of active ATM monomers that cross the nuclear membrane. If the flux of ATM monomers is not sufficient, some DSBs may not be recognized by NHEJ and therefore become unrepairable and lethal (leading to an increased radiosensitivity) and/or misrepaired by the MRE11-dependent recombination-like pathway (leading to an increased radiosusceptibility) [71]. If the radiation dose increases a bit more to reach the dose required for a sufficient flux of active ATM monomers in the nucleus, cell survival paradoxically increases, because more DSBs are recognized. Such a phenomenon describes a U-shaped survival curve in the range of 1 to 500 mGy. We have shown that this situation is quantitatively predictable by the RIANS model. For the first time, to our knowledge, the RIANS model provides a biologically-relevant explanation of the HRS phenomenon [71] (Figure 3). 

Altogether, these data show the importance of the ratio between the number of RI DSB and the number of active ATM monomers that diffuse in the nucleus. The dose d_HRS_, at which HRS is maximal, belongs to the range of 0.1–0.2 Gy for human non-transformed fibroblasts. Interestingly, d_HRS_ was shown to correspond to a minimal flux of ATM monomers to recognize DSB, and such value is preferentially reached in group II cells because of the delayed RIANS that characterizes these cells [71]. 

### 5.3. A Relevant Explanation for the Adaptive Response?

The adaptive response (AR) results in a protective phenomenon occurring after two successive doses: a first “priming” dose (d_AR_) precedes a period of time (Δt_AR_) and a “challenging” dose (D_AR_). The AR phenomenon occurs when the effect of d_AR_ + Δt_AR_ + D_AR_ is lower than that of D_AR_. The priming dose d_AR_ is generally interpreted as a stimulus of the cellular defenses to reply to the challenging dose [86,87,88]. However, despite a number of observations on different materials [88], the nature of such defenses is still unidentified.

HRS is a single-dose phenomenon, while AR is an effect due to two successive doses. Both HRS and AR are preferentially observed in moderately radiosensitive (group II) cells but not in radioresistant and hyper-radiosensitive cells [89,90,91]. More precisely, a recent review has provided evidence that HRS positive cells are systematically AR positive but the contrary is not verified, since the dose d_AR_ is generally much smaller than d_HRS_ [92]. The RIANS model has provided a biologically relevant interpretation of AR (Figure 3): The d_AR_ dose triggers the production of DSB and ATM monomers that diffuse in the nucleus. At this stage, the irradiation conditions are similar to HRS;The period of time Δt_AR_ favors the accumulation of the ATM monomers in the nucleus. However, if Δt_AR_ is too long, the number of ATM monomers will be reduced because of the limited activity half-time of the ATM protein. If Δt_AR_ is too short, the accumulation of active ATM monomers in the nucleus will be reduced;The challenging D_AR_ dose triggers the production of a high number of ATM monomers. However, the excess of remaining ATM monomers induced by d_AR_ and still active in nucleus after d_AR_ + Δt_AR_ will facilitate the biological response to the RI DSB induced by D_AR_: the effect of d_AR_ + Δt_AR_ + D_AR_ is therefore lower than that of the challenging D_AR_ dose alone.

The radioresistant (group I) cells do not need such an excess of ATM monomers due to d_AR_, since the flux of ATM monomers induced directly by D_AR_ is largely sufficient to provide a positive biological response. In hyper-radiosensitive (group III) cells, the flux of monomers is already so reduced that even an excess of ATM monomers induced by d_AR_ is not sufficient to provide a positive response to D_AR_. Hence, the RIANS model explains why AR is preferentially observed in moderately radiosensitive (group II) cells but not in group I and III cells. Interestingly, d_AR_ would correspond to the dose required for producing a sufficiently high flux of active ATM monomers to reduce the effect of the challenging dose D_AR_ [92] and therefore is likely to be similar to D_HRS_.

Lastly, with regard to the hormesis phenomenon, which is defined as a J-shaped dose-response curve and leads to a benefit to irradiated cells, some preliminary data suggest that at doses lower than 25 mGy, the flux of ATM monomers is very important, while this dose range corresponds to the absence of RI DSB. Hence, the excess of nuclear ATM kinase activity may contribute to reducing the spontaneous oxidative stress in the cells and contribute to decreasing genomic instability. However, such a phenomenon would be preferentially observed in radioresistant (group I) cells, since the spontaneous oxidative stress and the genomic instability of radiosensitive (group II) cells would be too high (experiments in progress).

### 5.4. Statins and Bisphosphonates: A New Approach of Radiological Protection?

In the group II cells, the radiosensitivity is explained by the delay in the RIANS. But what does happen if the RIANS is accelerated or facilitated? To date, the approach of radiological protection has consisted of decreasing the amount of RI DNA damage by using antioxidant drugs [93]. However, to decrease the amount of RI DNA damage does not necessarily mean that it will be repaired. Conversely, in the frame of the RIANS model, the diffusion of the ATM monomers depends on the permeability of the nuclear membrane [71]. Zoledronate is an anti-osteoporosis bisphosphonate agent and pravastatin is an anti-cholesterol drug. The combination of zoledronate and pravastatin (ZOPRA) was shown to inhibit the nuclear membrane farnesylation of progerin, a mutant form of nuclear lamin A, and to correct some biological features of cells from the progeroid Hutchinson–Gilford syndrome [33]. It is also likely that pravastatin may act on the cholesterol forms situated at the proximity of the nuclear membranes [94]. The application of ZOPRA treatment to fibroblasts was shown to significantly accelerate the rate of the RIANS, notably in cells derived from the Huntington’s disease [34], neurofibromatosis type I [44], tuberous sclerosis [45], Bruton’s [46] and Usher’s syndromes (data submitted), and from the progeroid Hutchinson–Gilford syndrome [33]. It is noteworthy that the significant effects of ZOPRA treatment is a crucial element that consolidates the RIANS model and makes it possible to open new approaches of radiological protection in mammalian cells (Figure 3).

## 6. Other Applications of the RIANS Model

To consolidate it, the biological relevance of the RIANS model should be tested in all the experimental conditions in which radiosensitivity varies. This is notably the case with the relationship between high linear energy transfer (LET) particles and the relative biology efficiency (RBE), that is a classical feature of radiobiology. Recently, the RIANS model was shown to explain the shape of the LET-RBE curves, with different types of particles like protons and carbon ions [74]. Similarly, the relevance of the RIANS model was also tested in dose hypo/hyper fractionation effects and on different radiotherapy modalities (experiments in progress). Furthermore, one of the important questions raised by radiobiology is the tissue-dependence of the response to radiation: an increasing body of evidence suggests that the RIANS is not specific to human fibroblasts and is easily observed in any mammalians cells, whether normal or tumor (paper in preparation). 

## 7. Conclusions

To predict individual radiosensitivity requires a number of constraints and the resolution of some historical enigmas of radiobiology. The current predictive assays do not reach all these requirements. In 2016, from a collection of more than 100 human primary fibroblasts cell lines, it was hypothesized that ionizing radiation induces the monomerization of cytoplasmic ATM dimers and triggers their diffusion in the nucleus to recognize DSBs and repair them via NHEJ [35,71]. This general mechanistic model makes it possible to describe most of the RI radiosensitivity-related phenomena, whether they occur after high- or low-dose. Obviously, as *primum movens* upstream of the DNA damage repair step, the RIANS does not explain by itself why, with the same delay in the RIANS, some syndromes are associated with a radiosensitivity + radiosusceptiblity or a radiosensitivity + radiodegeneration phenotype. However, some encouraging series of data have provided important elements on cancer proneness syndromes via the MRE11-recombination-like pathway, which is overactivated if RIANS is delayed [40], and on accelerated aging syndromes via the ATM monomers trafficking close to the nuclear membranes [33]. The RIANS model has also the considerable advantage that each of its steps is already described via mathematical modeling [71] and is opening the door to a new radiological protection approach with ZOPRA treatment. Further investigations are therefore needed to increase the spectrum of the biological and clinical applications of the RIANS model, in order to propose a unified view of the individual radiation response.

## Figures and Tables

**Figure 1 cancers-11-00905-f001:**
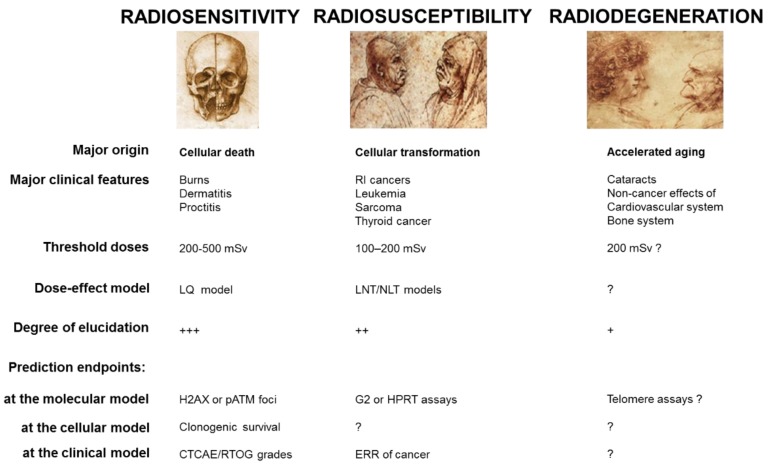
The major radiation-induced (RI) effects and their specific features. This figure aims to summarize the current basic knowledge of radiosensitivity, radiosusceptibility, and radiodegeneration. The threshold doses were reviewed in [2].

**Figure 2 cancers-11-00905-f002:**
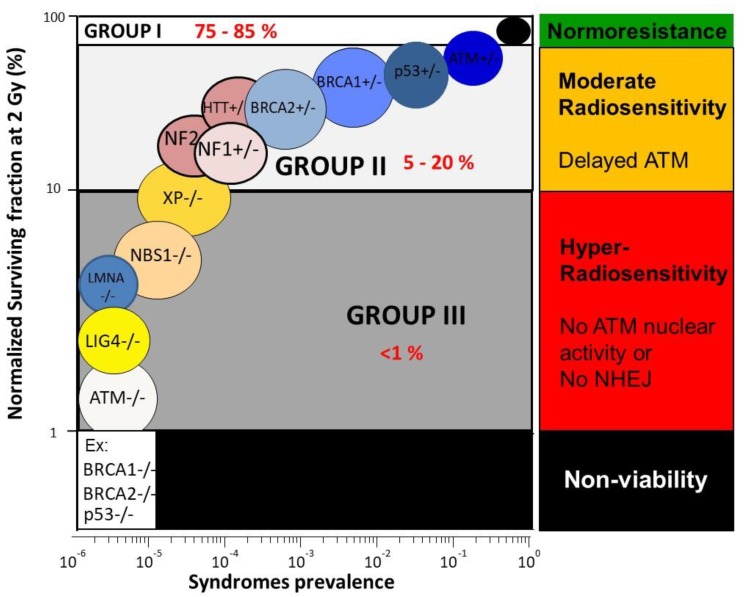
Cellular radiosensitivity as a function of syndrome prevalence. Survival fraction at 2 Gy (SF2) was fixed at 1% for ataxia telangiectasia mutated (ATM)-mutated cells and 100% for normosensitive patients. Each syndrome is represented by confidence zones. Data were taken from the databank of our lab and from [39].

**Figure 3 cancers-11-00905-f003:**
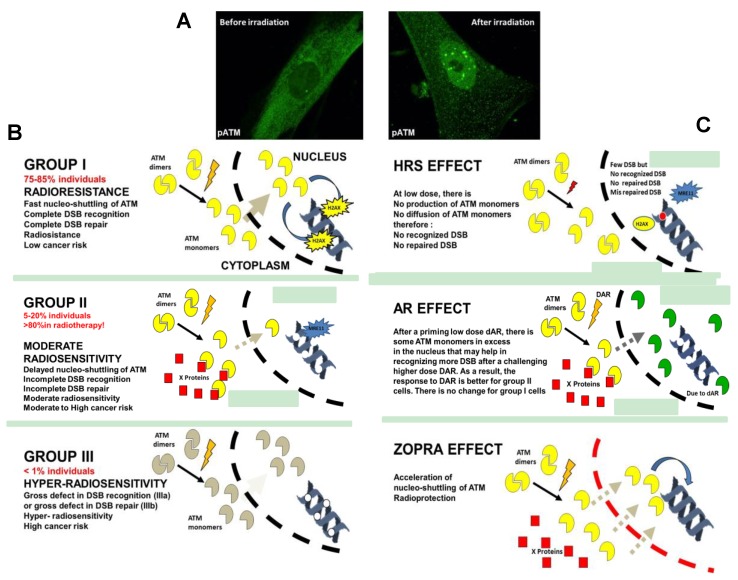
The radiation-induced ATM nucleoshuttling (RIANS) model and its applications. (**A**) Representative image of pATM immunofluorescence before or after irradiation (2 Gy) in human normosensitive control fibroblast cell lines. (**B**) Schematic illustration of the three groups of radiosensitivity defined from the RIANS model. (**C**) Schematic illustration of the hyperradiosensitivity to low doses (HRS) and the adaptive response (AR) phenomena and of the effect of the combination of statins and bisphosphonates (zoledronate+pravastatin, (ZOPRA)) on the RIANS. Since the link between radiosensitivity and the nuclear membrane permeability is still unknown, membranes are represented in the same manner.

**Table 1 cancers-11-00905-t001:** Numerical values of the major endpoints reflecting clinical, cellular, and molecular radiosensitivity ^1^.

Radiosensitivity of the Patients	CTCAE/RTOG Grade	SF2 (%)	γH2AX Foci at 10 min Post-Irradiation	pATM Foci at 10 min Post-Irradiation	γH2AX foci at 24 h Post-Irradiation
Group I	0	50–70	70–80	30–40	0–2
Group II	0–4	10–50	10–70	10–30	2–8
Group III	5	1–10	IIIa^2^: 0–5 IIIb^2^: 70–80	IIIa: 0 IIIb: 30–40	IIIa: 0–5 IIIb: 30–40

^1^ Experimental values were taken from [35]. The subgroup IIIa gathers syndromes with gross defects in the DSB recognition step like those caused by ATM mutations; the subgroup IIIb gathers syndromes with gross defects in the DSB joining step like those caused by LIG4 mutations [35].

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
