# Peer review of "The Nucleoshuttling of the ATM Protein: A Unified Model to Describe the Individual Response to High- and Low-Dose of Radiation?"

_cancers, 2019, doi:10.3390/cancers11070905_

Round 1

Reviewer 1 Report

This review briefly summarized factors affecting radiosensitivity, highlighted the critical role of ATM actions in the process, and focused on presenting the message that the ability of cytoplasm to nucleus trafficking of ATM is a unified model to predict individuals’ radiosensitivity.

1) Although this review does not have a dense content, it is rather difficult to follow. This is likely attributable to excessive use of the bulletin style writing. The style does not provide smooth progressions between sections, and requires readers to consistently keep in mind what the purposes of individual bulletin paragraphs were serving for. Additionally, individual bulletin paragraphs in the same group may not serve the same goal. For example, the first 3 bulletins of “Introduction” were for the clinical outcomes of radiation but the following four were for a different purpose. This arrangement goes throughout the manuscript.

Secondly, there was excessive concept or hypothesis development throughout the manuscript, for instance sections 2.1, 2.3, 2.4, 5.1, and others. The relevant information could be better organized or articulated to guild readers through concept or knowledge development.

2) Section 2 is simply too long for its content; it should be cut down substantially. 

3) While the review started as saying: “The evaluation of the radiation-induced (RI) risks is of medical, scientific and societal interest”, I couldn’t grad the importance of RI risk stratification in the current society from this review. It’s importance to which population of society, which cancer therapy, and localized radiation therapy should be discussed.

4) Some details of the major citations and concepts should be provided. For example, how ATM nucleoshuttling is evaluated; what are the underlying mechanisms; what is the evidence for monomer but not dimer ATM being shuttled; and is phosphorylation at S1981 required? Is it specific for radiation-induced DNA damage response? This knowledge is important, as pATM nuclear foci were used as the surrogate marker for the shuttle in this review. There must be numerous factors involved from the shuttle to the focus formation.

Details rather than giving simple statements supporting the ATM shuttle as an effective biomarker to classify RI risk should be given; for example the number of studies, cohort size, and more.

5) Gamma-H2AX is also produced by ATR and DNAPK. In comparison to ATM, DNAPK is more directly relevant to NHEJ. Are ATR and DNAPK relevant in predicting RI radiosensitivity?

6) One way for cell to choose NHEJ over HR is recruiting 53BP1 to DSBs. Is the number of 53BP1 nuclear foci predicting RI radiosensitivity?

7) Minors

Page 3, bulletin 1 – “… … functional assays … …”. What were these assays referred to? Bulletin 3 – “… … unrepaired, misrepaired, and cumulated DNA breaks, respectively” Are these specific for radiosensitivity, radiosusceptibility, and radiodegeneration? For example, can misrepaired DNA breaks contribute to other two effects and vice versa?

Page 6, section 2.4, bulletin 1 – “Assays based on apoptosis only are irrelevant for fibroblasts that do not show this type of cell death, and show irrelevant inversed correlation with clinical radiosensitivity” Not clear, please rephrase it.

Page 9, section 4.2 – “From the same COPERNIC fibroblasts collection, a quantitative correlation between CTCAE/RTOG severity grades (reflecting clinical radiosensitivity) and the number of the pATM foci (reflecting cellular radiosensitivity) [34].” Is this a complete sentence?

Author Response

REVIEWER 1

We thank the reviewer for his/her comments. The text has been deeply modified to reach all the requirements of the reviewers and more than 20 new references have been added to document each statement or hypothesis.

This review briefly summarized factors affecting radiosensitivity, highlighted the critical role of ATM actions in the process, and focused on presenting the message that the ability of cytoplasm to nucleus trafficking of ATM is a unified model to predict individuals’ radiosensitivity.

1) Although this review does not have a dense content, it is rather difficult to follow. This is likely attributable to excessive use of the bulletin style writing. The style does not provide smooth progressions between sections, and requires readers to consistently keep in mind what the purposes of individual bulletin paragraphs were serving for. Additionally, individual bulletin paragraphs in the same group may not serve the same goal. For example, the first 3 bulletins of “Introduction” were for the clinical outcomes of radiation but the following four were for a different purpose. This arrangement goes throughout the manuscript.

 OK We agree. Initially, we have deliberately chosen the bulletin style writing for clarity and it is noteworthy that the 2 other reviewers do not require any change vis-à-vis this style. However, to reach the requirements of the reviewer, we have endeavoured deleting the half of bulletin style writing paragraphs, notably the last half of introduction that has been deeply modified. See comments below.

Secondly, there was excessive concept or hypothesis development throughout the manuscript, for instance sections 2.1, 2.3, 2.4, 5.1, and others. The relevant information could be better organized or articulated to guild readers through concept or knowledge development.

 OK. We agree, notably for sections 2 (see below), in which the text has been reduced drastically. We have deleted some large parts of text. However, the section 5.1 contains the description of the LQ model that is the basis of radiobiology. The RIANS model is the one that permits the biological interpretation of the LQ model since the 70s. We strongly believe that the link between the RIANS model and the LQ model is essential to consider the RIANS model as mechanistically relevant. For this reason and since the 2 other reviewers did not ask to supress this part, we have kept it.

2) Section 2 is simply too long for its content; it should be cut down substantially. 

 OK. Notwithstanding the length of the section 2, we wanted to insist on the series of requirements that an idealistic predictive assay should reach to be reliable and biologically relevant. However, we agree that some parts maybe more technical and difficult to follow. We have deleted the former section 2.1, kept the former section 2.2 (now 2.1) that defines radiosensitivity and dispatched 2.3 in other sections. However, the section 2.4 reviewed briefly all the predictive approaches and it may be useful for the reader to know what are the advantages/inconvenients of each approach before detailing the RIANS model.

3) While the review started as saying: “The evaluation of the radiation-induced (RI) risks is of medical, scientific and societal interest”, I couldn’t grad the importance of RI risk stratification in the current society from this review. It’s importance to which population of society, which cancer therapy, and localized radiation therapy should be discussed.

OK We agree. Since the societal interest in predicting radiosensitivity is not directly the scope of the paper, we modified the introduction by adding a new chapter that gives three major examples of the importance of better evaluating the RI risk. They correspond to current questions for radiation oncologists, radiologists and specialists of radiological protection and we added new references for each example. see modified text page 1 lines 28-35

4) Some details of the major citations and concepts should be provided. For example, how ATM nucleoshuttling is evaluated;

OK. See modified text page 6 end of section 3.1 and page 9 end of section 4.2

what are the underlying mechanisms; what is the evidence for monomer but not dimer ATM being shuttled; and is phosphorylation at S1981 required?

OK. We agree : See modified text notably at the new section 4.1 page 8.

Is it specific for radiation-induced DNA damage response? This knowledge is important, as pATM nuclear foci were used as the surrogate marker for the shuttle in this review. There must be numerous factors involved from the shuttle to the focus formation.

OK. See our reply to the comment 5).

Details rather than giving simple statements supporting the ATM shuttle as an effective biomarker to classify RI risk should be given; for example the number of studies, cohort size, and more.

 OK. The RIANS model was frits built on the COPERNIC collection (with more than 100 fibroblast cell lines representing one of the largest spectrum of radiosensitivity). Some data from the different genetic syndrome cited in the text were thereafter added and mentioned in the text. See modified text  in all the body of the manuscript. See notably the beginning of section 4.1 page 8.

5) Gamma-H2AX is also produced by ATR and DNAPK. In comparison to ATM, DNAPK is more directly relevant to NHEJ. Are ATR and DNAPK relevant in predicting RI radiosensitivity?

OK You are right. See the new paragraph written about these two kinases page 6 new section 3.2

6) One way for cell to choose NHEJ over HR is recruiting 53BP1 to DSBs. Is the number of 53BP1 nuclear foci predicting RI radiosensitivity?

OK You are right to mention 53BP1. See the new paragraph written about these two kinases page 7 section 3.2

7) Minors

Page 3, bulletin 1 – “… … functional assays … …”. What were these assays referred to? Bulletin 3 – “… … unrepaired, misrepaired, and cumulated DNA breaks, respectively” Are these specific for radiosensitivity, radiosusceptibility, and radiodegeneration? For example, can misrepaired DNA breaks contribute to other two effects and vice versa?

This part has been deleted to reach the above requirement

Page 6, section 2.4, bulletin 1 – “Assays based on apoptosis only are irrelevant for fibroblasts that do not show this type of cell death, and show irrelevant inversed correlation with clinical radiosensitivity” Not clear, please rephrase it.

OK. We Agree. See modified text that has been deeply modified in section 2

Page 9, section 4.2 – “From the same COPERNIC fibroblasts collection, a quantitative correlation between CTCAE/RTOG severity grades (reflecting clinical radiosensitivity) and the number of the pATM foci (reflecting cellular radiosensitivity) [34].” Is this a complete sentence?

OK. We Agree. See modified text .

Reviewer 2 Report

This manuscript is a review of the ATM protein in radiation. One suggestion is that the authors may also mention the studies of ATM in medical oncology. For example, the authors may cite the references such as

Petersen, Lars F., et al. "Loss of tumour-specific ATM protein expression is an independent prognostic factor in early resected NSCLC." Oncotarget 8.24 (2017): 38326.

Author Response

REVIEWER 2

We thank the reviewer for his/her comments. The text has been deeply modified to reach all the requirements of the reviewers and more than 20 new references have been added to document each statement or hypothesis.

This manuscript is a review of the ATM protein in radiation. One suggestion is that the authors may also mention the studies of ATM in medical oncology. For example, the authors may cite the references such as

Petersen, Lars F., et al. "Loss of tumour-specific ATM protein expression is an independent prognostic factor in early resected NSCLC." Oncotarget 8.24 (2017): 38326.

OK. We agree : See modified text notably at the new section 4.1 page 8.

Reviewer 3 Report

The manuscript “The nucleoshuttling of the ATM protein: a unified model to describe the individual response to high- and low-dose of radiation?” reviews the types of RI responses and propose a mechanism/assay to predict these responses with ATM. ATM is central in DNA repair of double stranded breaks induced by radiation. The authors argue ATM nucleoshuttling as a better/more reliable measure of radiosensitivity than cell death, cytogenic, DNA repair or genomic assays, particularly when identifying the types of radiosensitivity. The authors also speculate on the role in different phenomena such as hypersensitivity at low doses and adaptive response.

The reviewer’s comments are included below. Minor comments have also been included directly on the manuscript.

The authors need to be sure to include appropriate references for factual statements.

-Define ATM abbreviation at first appearance.

-In the first sentence “Less than 10 years after the X-rays discovery by Roentgen [1], the three major clinical 28 consequences of an irradiation were identified already…”

This sentence implies the definitions of radiosensitivity, radiosusceptibility and radiodegeneration were identified and widely accepted around 1905. However, these terms were proposed by the authors themselves (Foray et al 2016; Britel et al 2018), and should be referenced as such.

-Pg 3

The authors state common points for RI effects, however, do not discuss dose rate. Dose rate has been established as important influencer of radiation induced response (Brooks et al. 2016, for example).

-Pg 3 Line 72-74

“Interestingly, the radiation doses above which the radiosensitivity reactions, the RI cancers and the RI cataracts occur are of the same range (between 0.1 to 0.5 Gy). How to explain these common dose range?

It is unclear what the authors are stating. That RI cancers and cataracts occur above 0.5Gy? Why include a dose range and why would this be in question? Also is this dose referring to whole body or tissue specific exposure? References should be included for dose(s) at which RI cancers and cataracts occur.

-Pg 3 Line 79-83

Individual- and tissue-dependence specificities: the occurrence of the three precited RI effects strongly depends on the individual status and on the irradiated tissues/organs. The prediction of individual radiation response therefore requires a wide spectrum of individuals/tissues to be tested. How to predict the contribution of the individual factor in the response to radiation? How to take into account the tissue-dependence?”

Should be noted that considerations have been made to predict tissue radiosensitivity, such as the cell turnover time for specific tissue (as reviewed by Brooks et al. 2016).

-What is the source of the threshold doses in Figure 1?

-Pg 11 Line 389-390

“At low doses, the number of RI DSB and ATM monomers is reduced…”

The authors need to provide a reference for this statement. If this is theorized by the authors then it needs to be written as such.

-Pg 12 Line 396-397

“Such phenomenon describes a U-shaped survival curve in the range of 1 to 500 mGy (Figure 4).”

This sentence is misleading, as Figure 4 does not show a U-shaped curve or include a dose range from 1 to 500mGy.

-Pg 12 Line 439-440

“Conversely, in the frame of the RIANS model, the 439 diffusion of the ATM monomers depends on the permeability of the nuclear membrane”

The paper would benefit from mentioning the importance of nuclear membrane permeability earlier. In Figure 4 it appears the authors are suggesting in Group II and AR effect there is decreased nuclear membrane permeability.

-Hyper radiosensitivity, moderate radiosensitivity and radioresistence response at of human non-transformed fibroblasts at SF2 are discussed at length. As such, when discussion the hyper radiosensitivity seen at low dose or adaptive response, it should be stated if these phenomena have been observed also in human non-transformed fibroblasts, or in other cell types.

-The paper would benefit from the inclusion of references on the adaptive response discussed on page 6. One possibility is Feinendegen 2016, which provides an analysis of 18 studies on adaptive response. This paper also includes incidences of radiosensitivity at very low doses, lower than discussed by the authors.

-What are the author’s thoughts on nucleoshuttling of ATM in radiation hormesis? Do the authors believe this response is seen in those that can be categorized in Group 1 (Figure 4)? Or does hormesis belong in another response model not yet theorized by the authors?

Author Response

REVIEWER 3

We thank the reviewer for his/her comments. The text has been deeply modified to reach all the requirements of the reviewers and more than 20 new references have been added to document each statement or hypothesis.

The manuscript “The nucleoshuttling of the ATM protein: a unified model to describe the individual response to high- and low-dose of radiation?” reviews the types of RI responses and propose a mechanism/assay to predict these responses with ATM. ATM is central in DNA repair of double stranded breaks induced by radiation. The authors argue ATM nucleoshuttling as a better/more reliable measure of radiosensitivity than cell death, cytogenic, DNA repair or genomic assays, particularly when identifying the types of radiosensitivity. The authors also speculate on the role in different phenomena such as hypersensitivity at low doses and adaptive response.

The reviewer’s comments are included below. Minor comments have also been included directly on the manuscript.

OK See modified text as required by reviewer in the PDF file associated with this review

The authors need to be sure to include appropriate references for factual statements.

OK you are right. We have added new references accordingly.

-Define ATM abbreviation at first appearance.

 OK See modified text in abstract and at the end of page 3

-In the first sentence “Less than 10 years after the X-rays discovery by Roentgen [1], the three major clinical 28 consequences of an irradiation were identified already…”

This sentence implies the definitions of radiosensitivity, radiosusceptibility and radiodegeneration were identified and widely accepted around 1905. However, these terms were proposed by the authors themselves (Foray et al 2016; Britel et al 2018), and should be referenced as such.

 OK. In fact, in the text, we mentioned the fact that the RI consequences have been described but which does not necessarily imply that the terms describing the RI effects were fixed. We have therefore added at the end of each paragraph the explanation about the term used or that should be used to date. See modified text page 2

-Pg 3

The authors state common points for RI effects, however, do not discuss dose rate. Dose rate has been established as important influencer of radiation induced response (Brooks et al. 2016, for example).

 OK We agree. See modified text and added references at the end of the page 2.

-Pg 3 Line 72-74

“Interestingly, the radiation doses above which the radiosensitivity reactions, the RI cancers and the RI cataracts occur are of the same range (between 0.1 to 0.5 Gy). How to explain these common dose range?”

It is unclear what the authors are stating. That RI cancers and cataracts occur above 0.5Gy? Why include a dose range and why would this be in question? Also is this dose referring to whole body or tissue specific exposure? References should be included for dose(s) at which RI cancers and cataracts occur.

 OK we agree. Anyway, this paragraph as deleted to reach the requirement of the reviewer 1.

-Pg 3 Line 79-83

Individual- and tissue-dependence specificities: the occurrence of the three precited RI effects strongly depends on the individual status and on the irradiated tissues/organs. The prediction of individual radiation response therefore requires a wide spectrum of individuals/tissues to be tested. How to predict the contribution of the individual factor in the response to radiation? How to take into account the tissue-dependence?”

Should be noted that considerations have been made to predict tissue radiosensitivity, such as the cell turnover time for specific tissue (as reviewed by Brooks et al. 2016).

 OK. We agree : See modified text notably at the new section 4.1 page 8.

-What is the source of the threshold doses in Figure 1?

 OK for convenience, we gather all the references under the reference 2 that is a general review. See modified legend.

-Pg 11 Line 389-390

“At low doses, the number of RI DSB and ATM monomers is reduced…”

The authors need to provide a reference for this statement. If this is theorized by the authors then it needs to be written as such.

 OK you are right. See modified text section 5.2 page 10

-Pg 12 Line 396-397

“Such phenomenon describes a U-shaped survival curve in the range of 1 to 500 mGy (Figure 4).”

This sentence is misleading, as Figure 4 does not show a U-shaped curve or include a dose range from 1 to 500mGy.

 OK. You are right. The citation of Figure 4 is wrong. We have deleted this citation. See modified text page 10.

-Pg 12 Line 439-440

“Conversely, in the frame of the RIANS model, the 439 diffusion of the ATM monomers depends on the permeability of the nuclear membrane”

The paper would benefit from mentioning the importance of nuclear membrane permeability earlier.

OK You are right. However, the notion of permeability of nuclear membrane and its influence in radiosensitivity has never been raised before the hypotheses of the RIANS model since all the causes of radiosensitivity were believed to be restricted to nucleus and DSB repair. It was therefore logical to evoke it only from this step. We have mentioned it at the end of page 4.

In Figure 4 it appears the authors are suggesting in Group II and AR effect there is decreased nuclear membrane permeability.

OK you are right. In fact we could hypothesize that membrane in Group II is more obturated or less permeable to ATM monomers. We have preferred to change the figure and to evoke this assumption in the text. See modified figure and text page 8.

 OK. You are right.

-Hyper radiosensitivity, moderate radiosensitivity and radioresistence response at of human non-transformed fibroblasts at SF2 are discussed at length. As such, when discussion the hyper radiosensitivity seen at low dose or adaptive response, it should be stated if these phenomena have been observed also in human non-transformed fibroblasts, or in other cell types.

  OK you are right. See modified text section 5.2 page 10

-The paper would benefit from the inclusion of references on the adaptive response discussed on page 6. One possibility is Feinendegen 2016, which provides an analysis of 18 studies on adaptive response. This paper also includes incidences of radiosensitivity at very low doses, lower than discussed by the authors.

   OK you are right. See modified text section and additional reference in section 5.3 page 11

-What are the author’s thoughts on nucleoshuttling of ATM in radiation hormesis? Do the authors believe this response is seen in those that can be categorized in Group 1 (Figure 4)? Or does hormesis belong in another response model not yet theorized by the authors?

OK . Very interesting question but a paper is in preparation about RIANS and hormesis. See modified text in the end of Section 5.3 page 12.

Round 2

Reviewer 1 Report

The revision has sufficiently addressed my comments and is much improved. The review addresses a topic that has yet to be thoroughly investigated, and contributes to the literature in this field.

Author Response

We thank the reviewer.

Reviewer 2 Report

All of my concerns are addressed. The manuscript can be accepted.

Author Response

We thank the reviewer

Reviewer 3 Report

Minor comments have been made directly to the pdf.

Author Response

We thank the reviewer. 

All the corrections proposed by the reviewer in the PDF file have been addressed.